# Virtual reality environment using a dome screen for procedural pain in young children during intravenous placement: A pilot randomized controlled trial

Ha Ni Lee[1], Woori Bae[2], Joong Wan Park[1]*, Jae Yun Jung[1], Soyun Hwang[1], Do Kyun Kim[1], Young Ho Kwak[1]

1 Department of Emergency Medicine, Seoul National University Hospital, Seoul, Republic of Korea,
2 Department of Emergency Medicine, Seoul St. Mary Hospital, College of Medicine, The Catholic University of Korea, Seoul, Republic of Korea

☉ These authors contributed equally to this work.
* zzibii@naver.com

**Data Availability Statement:** All relevant data are within its Supporting Information files.

## Abstract

We assessed the feasibility and potential efficacy of a virtual reality (VR) environment using a dome screen as a distraction method in young children during intravenous (IV) placement in the pediatric emergency department. This randomized controlled pilot study enrolled children aged 2 to 6 years who underwent IV placement into either the intervention group or the control group. Children in the intervention group experienced VR using a dome screen during IV placement. The child's pain intensity was measured using the Face, Legs, Activity, Cry, and Consolability (FLACC) scale at four time points of IV placement: immediately after arrival to the blood collection room (base); immediately after the child laid down on the bed (preparation); when the tourniquet was applied (tourniquet); and the moment at which the needle penetrated the skin (venipuncture). The guardian's satisfaction and rating of the child's distress were assessed using a 5-point Likert-type questionnaire. We recruited 19 children (9 in the intervention group and 10 in the control group). Five children in the control group were excluded from the analysis because of missing video recordings (n = 3), failed first attempt at IV placement (n = 1), and the child's refusal to lie on the bed during the procedure (n = 1). No side effects of VR were reported during the study period. Although the average FLACC scale score at each time point (preparation, tourniquet, venipuncture) was lower in the intervention group than the control group, the difference was not statistically significant (2.3, interquartile range [IQR]: 2.0–3.0; vs. 3.3, IQR: 2.7–6.7, $P = 0.255$). There were no statistically significant differences between the groups in the guardian's satisfaction and anxiety or his/her rating of the child's pain and anxiety. The guardians and emergency medical technicians reported satisfaction with the use of VR with a dome screen and considered it a useful distraction during the procedure. VR using a dome screen is a feasible distraction method for young children during IV placement. A larger clinical trial with further development of the VR environment and study process is required to adequately evaluate the efficacy of VR using a dome screen.

**Funding:** This work was supported by the National Research Foundation of Korea (grant no. 2019R1C1C1011000) and Seoul National University Hospital (grant no. 04-2018-0760). The sponsors were not involved in the study design; data collection, analysis or interpretation; writing of the manuscript; or the decision to submit the manuscript for publication.

**Competing interests:** The authors have declared that no competing interests exist.

# Introduction

Visiting the pediatric emergency department (PED) is severely stressful for children and their guardians because most children presenting to the PED are in pain or require painful procedures [1]. Needle procedures, such as venipuncture and intravenous (IV) placement, are the most common causes of pain in hospitalized children [2]. IV placement is the most common invasive procedure performed in PEDs [3–5]. These pain- and anxiety-causing procedures may cause psychological trauma for children, make the treatment process difficult and worsen the relationship between the guardian and medical staff [6].

Appropriate pain control is essential for improving patient management, and various analgesics and nonpharmacological strategies were studied to control pain in children in the PED [7]. Among the nonpharmacological methods, digital distraction was actively studied in recent years [8]. Virtual reality (VR), often referred to as a virtual environment, is a computer technology that enables users to view or 'immerse' themselves in an alternate world, and it is attracting attention as a digital distraction technique. VR provides a clinically important reduction in pain during various procedures, such as venipuncture, burn dressing, and lumbar puncture [9–16].

However, previous studies generally tested VR in school-aged children. Commercially available VR systems generally require the wearing of a head-mounted display (HMD) helmet to create a virtual space and block out the real world, which may be challenging for young children [17]. The primary age group int the PED is children younger than 6 years of age, and it is essential to test VR distraction in this age group to demonstrate its effectiveness for pain reduction in the PED [18].

Since wearing an HMD is difficult for young children, we developed a new approach of delivering VR via a dome screen. To the best of our knowledge, no study has tested whether VR distraction using a dome screen is helpful for significant pain reduction during needle procedures for young children ($\leq$ 6 years old) in the PED. The current pilot randomized clinical trial assessed the feasibility and acceptability of VR distraction using a dome screen for young children during needle procedures in PEDs. The secondary aims were to obtain preliminary results on the efficacy of VR distraction using a dome screen and to determine the sample size needed in a future larger clinical trial.

# Methods

## Trial design

This study was a single-center, two-arm parallel, feasibility pilot randomized controlled trial with allocation by the date of PED visit. The institutional review board (IRB) of Seoul National University Hospital approved this study prior to its start date (IRB no. 1901-094-1005), and it was registered at cris.nih.go.kr (KCT0005691) after completion. Registration of this pilot trial was performed retrospectively because the authors were less aware of the required prospective registration for the pilot trial. The authors confirm that all ongoing and related trials for this intervention are registered.

## Participants

Children visiting the PED were eligible if they were aged 2 to 6 years and underwent IV placement for diagnosis or treatment. Children were excluded from the study if they needed urgent IV placement (e.g., due to an unstable hemodynamic condition or an altered mental status), had developmental disabilities or facial anomalies that made it difficult to use the pain scale, and if their guardians had insufficient Korean language ability to understand the study

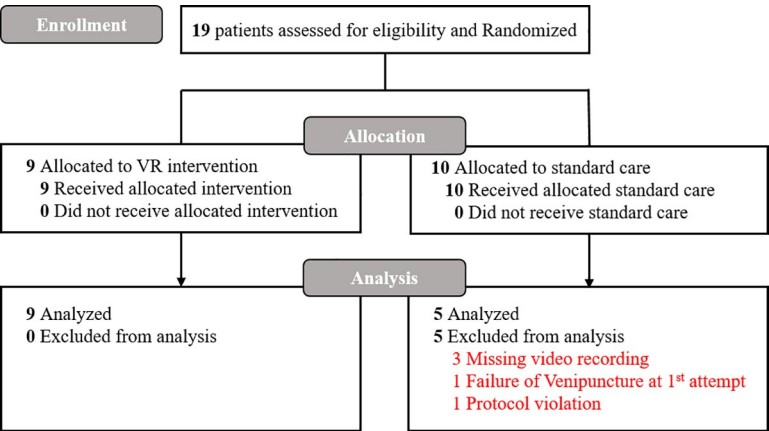

**Fig 1. Enrollment and randomization of participants in the study.**

protocol. Children for whom IV catheter insertion failed on the first attempt were also excluded from the study. Fig 1 shows a flowchart of this study.

This pilot randomized controlled trial was performed at the Seoul National University Children's Hospital (SNUCH) in Seoul, South Korea between May and September 2019. SNUCH is a tertiary academic hospital with a 315-bed capacity, and over 20,000 children visit its PED annually.

## Randomization and blinding

Children were randomized to either the intervention group or the control group according to the dates of their ED visits. Children whose ED visit date was an odd number were assigned to the intervention group with VR distraction, and children whose ED visit date was an even number were assigned to the control group. Allocation concealment was impossible for the trial staff recruiting participants in this randomization method. Blinding was not feasible due to the nature of the intervention.

## Interventions

**Study protocol.** The study protocol is illustrated in Fig 2. If the patient was eligible, trial staff explained the study and obtained written informed consent from his or her guardians. Five minutes before the emergency medical technician (EMT) started the IV placement, the child entered the blood collection room with his or her guardian, and the trial staff began video recording. By adjusting the angle of view, the dome screen was not visible in the recorded video to blind each participant's allocation to PED staff who rated the pain scale from the recorded video. For the children assigned to the intervention group, the VR animation was projected through the dome screen within 1 minute after the child entered the blood collection room. Children in the control group were asked to lie on a bed for IV placement without VR animation via a dome screen. The EMT began IV placement according to the following sequence: Tourniquet application; venipuncture site cleansing; venipuncture; and indwelling IV cannula insertion. No local anesthetics or analgesics other than the VR intervention were applied during the procedure. The guardian was allowed to hold the child's opposite arm for reassurance during the procedure. Video recording continued to approximately 2 minutes after venipuncture. After the participant left the blood collection room, the guardian received and completed a questionnaire about the needle procedure.

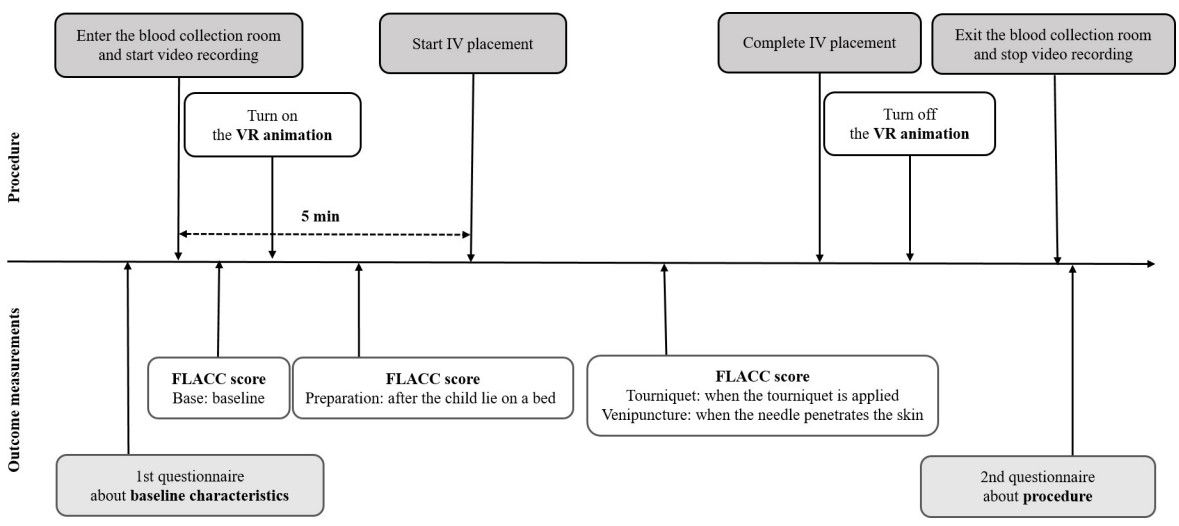

**Fig 2. Study protocol.**

**VR environment using a dome screen.** Our VR equipment consisted of a dome screen developed by Dome & Dome Co. and a projector (EB-G7100, EPSON, Japan) linked to a personal computer that played the animated show 'Pororo the Little Penguin' (Fig 3). The diameter and height of the dome screen were 1600 mm and 600 mm, respectively. The dome screen was placed at the end of the bed so the child could watch the screen while lying down during the procedure. We projected the animation onto the dome screen using the projection mapping software program MadMapper version 4.0 to provide the children with a VR environment [19]. The animation used in this study was a famous Korean animated show that primarily targets children aged 3 to 5 years. We used a free downloadable episode from the official 'Pororo the Little Penguin' channel on the online video sharing platform YouTube (https://www.youtube.com/watch?v=A5SZuwf0e98&t=172s). Before the start of the study period, we received approval from the production company for the use of the episode for research purposes.

## Outcomes

The primary outcome was the observed pain intensity during IV placement. Pain intensity was measured using the Face, Legs, Activity, Cry, Consolability (FLACC) scale at 4 time points during the needle procedure: immediately after arrival at the blood collection room (base); immediately after the child laid on the bed (preparation); when the tourniquet was applied (tourniquet); and when the needle penetrated the skin (venipuncture). The FLACC scale is a validated pain measure for children who cannot report pain. The 5 elements of the FLACC scale are each scored on a range from 0–2 then summed for a total score ranging from 0–10, with higher scores indicating greater pain intensity [20]. Two PED staff members who were blinded to the study protocol completed pain intensity assessments based on video recordings. The two PED staff members were blinded to each child's allocation by setting the volume in the video recordings of both groups to zero and independently observing the video recording of the needle procedure for each child.

The secondary outcomes included the guardian's satisfaction with the procedure and rating of their child's distress (pain, anxiety), which were assessed using a 5-point Likert-type questionnaire after the procedure. The feasibility and acceptability of this trial were assessed by asking PED staff and EMTs who participated in the needle procedure about their satisfaction with

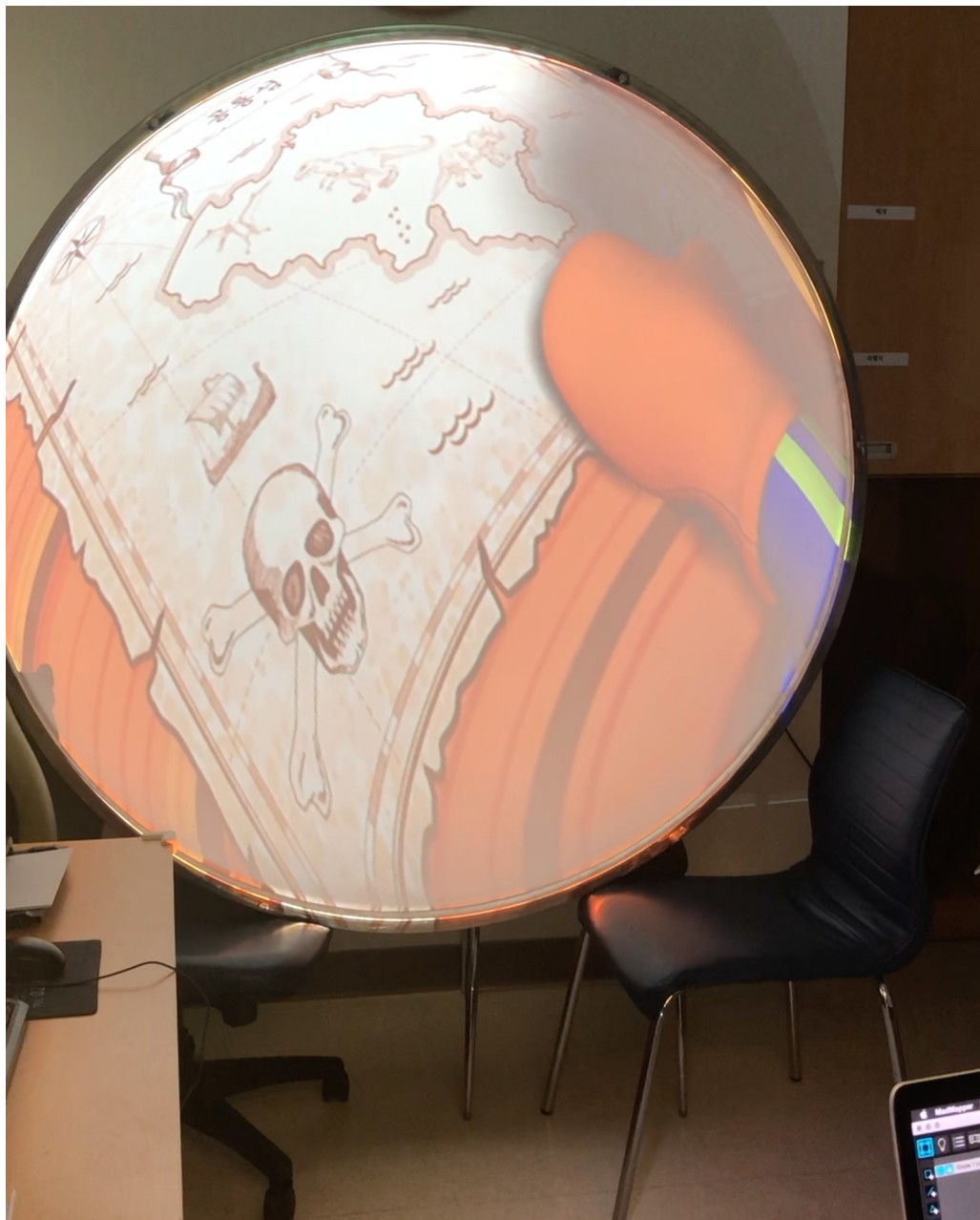

**Fig 3. Virtual reality environment using a dome screen in the blood collection room.**

the process of providing VR intervention to children as a distraction during the needle procedure. If they answered that they were unsatisfied with VR intervention and the study process, further details were documented. The perceived effect on the child's distress and side effects were also documented.

## Sample size

Similar to previous pilot studies on distraction methods for children during painful procedures, formal sample size calculations were not required [21–23]. We aimed to recruit

approximately 20 children to provide sufficient preliminary evidence of the clinical efficacy of VR as a distraction method.

## Statistical methods

The baseline variables are described using appropriate summary statistics. Categorical variables are reported as frequencies and percentages. Continuous variables are reported as medians and interquartile ranges (IQRs). The FLACC scale scores at each time point and the average of the FLACC scores at 3 time points (preparation, tourniquet, and venipuncture) were compared between the intervention group and control group using the Mann-Whitney U test and the independent t-test. A between-group comparison of the guardians' responses to the postprocedural questionnaire was performed using Fisher's exact test. All statistical tests were performed at a significance level of 0.05 (2-sided) using STATA version 14.2 (StataCorp LP, College Station, TX, USA).

## Results

During the recruitment periods, 19 children were eligible, and all of the eligible children's guardians consented to participate in this study (Fig 1). We enrolled these 19 children (9 in the intervention group and 10 in the control group). Of the 10 children allocated to the control group, five were excluded from data analysis because of missing video recordings (n = 3), failure of IV placement on the first attempt (n = 1), and refusal to lie down during the procedure (n = 1). The analysis was performed with a total sample of 14 children.

Table 1 provides information on the baseline characteristics of the participants. The overall median age was 4.5 (interquartile range [IQR]: 3.0–5.8) years, and 28.6% were boys. Approximately 71% of the children had previous experience with venipuncture, and 84.6% of their

**Table 1. Baseline characteristics of the participants.**

| Characteristic | Intervention | | Control | | Total | | P value |
|---|---|---|---|---|---|---|---|
| | (n = 5) | | (n = 9) | | (n = 14) | | |
| Age, median (IQR), years | 3 | (3.0–5.0) | 5 | (3.0–6.0) | 4.5 | (3.0–5.8) | 0.585 |
| Sex, male, n (%) | 0 | (0.0) | 4 | (44.4) | 4 | (28.6) | 0.221 |
| Reason for PED visit, n (%) | | | | | | | 0.258 |
| Disease | 5 | (100.0) | 6 | (66.7) | 11 | (78.6) | |
| Trauma | 0 | (0.0) | 3 | (33.3) | 3 | (21.4) | |
| Previous venipuncture, yes, n (%) | 4 | (80.0) | 6 | (66.7) | 10 | (71.4) | 1.000 |
| Analgesic medication in the past 2 h, yes, n (%) | 2 | (40.0) | 4 | (44.4) | 6 | (42.9) | 1.000 |
| Guardian's characteristics | | | | | | | |
| Age, median (IQR), years | 38 | (37.0–40.0) | 39 | (38.0–42.0) | 38 | (38.0–42.0) | 0.424 |
| Relationship, n (%) | | | | | | | 0.505 |
| Mother | 5 | (100.0) | 7 | (77.8) | 12 | (85.7) | |
| Father | 0 | (0.0) | 2 | (22.2) | 2 | (14.3) | |
| Number of children, median (IQR) | 1.0 | (1–2) | 2 | (1–3) | 1.5 | (1.0–2.6) | 0.903 |
| Previous observation of the child during a procedure, yes, n (%)[a] | 3 | (75.0) | 8 | (88.9) | 11 | (84.6) | 0.505 |

PED, pediatric emergency department; IQR, interquartile range.

[a] One child in the intervention group was excluded from the analysis because of missing data.

**Table 2. Pain assessed using the FLACC scale in the intervention and control groups.**

| | Group | | | | P value[a] |
|---|---|---|---|---|---|
| | **Intervention (n = 5)** | | **Control (n = 9)** | | |
| Base | | | | | |
| Mean (SD) | 0.0 | (0.0) | 0.0 | (0.0) | |
| Median (IQR) | 0.0 | (0.0–0.0) | 0.0 | (0.0–0.0) | |
| Range | 0.0 to 0.0 | | 0.0 to 0.0 | | |
| Preparation | | | | | 0.309 |
| Mean (SD) | 1.6 | (2.1) | 3.2 | (2.9) | |
| Median (IQR) | 1.0 | (0.0–2.0) | 3.0 | (1.0–6.0) | |
| Range | 0.0 to 5.0 | | 0.0 to 5.0 | | |
| Tourniquet | | | | | 0.139 |
| Mean (SD) | 2.2 | (1.8) | 4.7 | (3.3) | |
| Median (IQR) | 2.0 | (2.0–2.0) | 4.0 | (3.0–7.0) | |
| Range | 0.0 to 5.0 | | 0.0 to 5.0 | | |
| Venipuncture | | | | | 0.545 |
| Mean (SD) | 3.8 | (2.3) | 4.8 | (2.5) | |
| Median (IQR) | 3.0 | (3.0–5.0) | 4.0 | (3.0–8.0) | |
| Range | 1.0 to 7.0 | | 0.0 to 9.0 | | |
| Mean[b] | | | | | 0.255 |
| Mean (SD) | 2.5 | (1.7) | 4.2 | (3.1) | |
| Median (IQR) | 2.3 | (2.0–3.0) | 3.0 | (2.7–6.7) | |
| Range | 0.3 to 5.0 | | 0.0 to 8.7 | | |

FLACC, Face, Legs, Activity, Cry, Consolability; SD, standard variation; IQR, interquartile range.

[a] Mann-Whitney U test.

[b] Average scores at 3 time points (preparation, tourniquet, and venipuncture).

guardians had previously observed the child's venipuncture. There were no significant differences in baseline characteristics between the intervention and control groups.

Table 2 presents a comparison of the FLACC scale scores between the two groups. There was no significant difference in the FLACC scale at any time point during the needle procedure. The median FLACC scale at the time of venipuncture was 3.0 (IQR: 3.0–5.0) in the intervention group and 4.0 (IQR: 3.0–8.0) in the control group ($p = 0.545$). Although the average FLACC score at the 3 time points (preparation, tourniquet, venipuncture) in the intervention group was lower than the control group, the difference was not statistically significant (median 2.3 (IQR: 2.0–3.0) vs. 3.0 (IQR: 3.0 (IQR 2.7–6.7)), $p = 0.255$).

Based on these data in a post hoc sample size calculation (STATA version 14.2), we calculated that with 80% power and a 2-sided alpha of 0.05, a sample size of 36 per group would be required to detect possible subtle differences.

Table 3 summarizes the results of the guardians' responses to the postprocedural questionnaire. The guardians' ratings of the child's pain and anxiety differed between the groups, but the difference was not statistically significant ($p = 0.540$ and $p = 1.000$, respectively). Zero percent of the guardians in the intervention group felt that their child's experience was very painful compared with 22.2% in the control group. There were no statistically significant differences between the groups in guardian satisfaction with and anxiety for the overall procedure ($p = 1.000$ and $p = 0.830$, respectively).

**Table 3. Guardian's response to postprocedural questionnaires.**

| | Intervention (n = 5) | | Control (n = 9) | | *P* value |
|---|---|---|---|---|---|
| Guardian's rating of child's pain | | | | | |
| 1 (Very painful) | 0 | (0.0%) | 2 | (22.2%) | |
| 2 | 1 | (20.0%) | 0 | (0.0%) | |
| 3 | 2 | (40.0%) | 2 | (22.2%) | 0.540 |
| 4 | 2 | (40.0%) | 3 | (33.3%) | |
| 5 (Not painful) | 0 | (0.0%) | 2 | (22.2%) | |
| Guardian's rating of child's anxiety | | | | | |
| 1 (Very anxious) | 0 | (0.0%) | 2 | (22.2%) | |
| 2 | 1 | (20.0%) | 1 | (11.1%) | |
| 3 | 2 | (40.0%) | 3 | (33.3%) | 1.000 |
| 4 | 1 | (20.0%) | 2 | (22.2%) | |
| 5 (Not anxious) | 1 | (20.0%) | 1 | (11.1%) | |
| Guardian's anxiety during the procedure | | | | | |
| 1 (Very anxious) | 1 | (20.0%) | 1 | (11.1%) | |
| 2 | 1 | (20.0%) | 3 | (33.3%) | |
| 3 | 2 | (40.0%) | 4 | (44.4%) | 0.830 |
| 4 | 1 | (20.0%) | 0 | (0.0%) | |
| 5 (Not anxious) | 0 | (0.0%) | 1 | (11.1%) | |
| Guardian's satisfaction with the overall procedure | | | | | |
| 1 (Very dissatisfied) | 0 | (0.0%) | 0 | (0.0%) | |
| 2 | 0 | (0.0%) | 0 | (0.0%) | |
| 3 (Neutral) | 1 | (20.0%) | 1 | (11.1%) | 1.000 |
| 4 | 3 | (60.0%) | 6 | (66.6%) | |
| 5 (Very satisfied) | 1 | (20.0%) | 2 | (22.2%) | |

## Discussion

This pilot study showed that VR using a dome screen may be a feasible distraction method for young children during a needle procedure. It was easier than we expected to receive written consent from guardians to test the VR using a dome screen during the needle procedure. The guardians may have felt that VR using a dome screen was very unlikely to harm their child compared to a pharmacological distraction method. No side effects from VR were reported during the study period.

However, we were unable to recruit the planned number of children. Because the construction period of a PED expansion project overlapped with our study periods, we did not have adequate time to recruit children within the study periods. Approximately 15% of children were also excluded from the final analysis because of missing video recordings. Minor mistakes due to the use of unfamiliar equipment may explain the missing data, which may be reduced via sufficient practice and preparation before the start of the main study.

Because of the small number of participants, it was impossible to show a statistically meaningful efficacy of VR distraction using a dome screen. The differences in the FLACC scale between the groups were not statistically significant overall, but the children allocated to the intervention group showed a numerically smaller average score on the FLACC scale. The guardians in the intervention group also showed great satisfaction with VR using a dome screen as a distraction method in terms of its effects on their children's pain and anxiety.

There was also not sufficient time to allow the child to concentrate on the animated show before the needle procedure. We planned the study protocol for the children to watch the VR

animation for approximately 4 minutes before the EMTs entered the room, but the duration of time needed to watch the VR animation for a sufficient distraction effect was different for each child. It will be necessary to give children enough time to focus on the animated show before the needle procedure in future studies.

There were some technical issues in the implementation of VR using a dome screen in clinical practice. The child needed to tilt his or her head to watch the animated show to avoid having part of the dome screen hidden by their body or the bed. Although for the purposes of study standardization, the participants were required to lie on the bed, one child insisted on watching the animated show in a sitting position. It may be more appropriate in future studies to fix the dome screen to the ceiling or reposition it above the child's face. Turning off the lights in the room after the animated show started also made it more difficult for the EMTs to perform the needle procedure. Although portable light was provided temporarily, the EMTs reported that it was more difficult than normal to find blood vessels to puncture.

The present study has some limitations. First, participants were randomized according to the date of their ED visit in this pilot study. We used this method for easy implementation. However, it may lead to unbalanced groups due to the high risk of selection bias. We should consider more appropriate randomization methods, such as permuted block randomization, in a future larger clinical trial to balance sample size across groups and minimize bias. Second, it was impossible to blind the children, guardians, EMTs and PED staff due to the nature of the intervention. PED staff members' knowledge of which children were assigned to the intervention group may have affected the study process and potentially led to biased results. For the PED staff to observe the video recording, the light from the dome screen prevented blinding to whether the child was assigned to the intervention group. Using a close-up shot of the children's face and body may help blind the PED staff reviewing video recordings to whether the animated program was playing in the room during the procedure. Third, we did not consider the effect of covariates, such as the indication for IV placement and the child's previous experience with IV placement, on baseline pain intensity and anxiety. Fortunately, all of the children's baseline FLACC scales were zero despite differences in baseline characteristics. Children with traumatic injuries may have greater baseline pain intensity and anxiety than children with medical diseases. Possible covariates that may affect baseline pain intensity should be considered in future studies.

## Conclusion

VR using a dome screen is a feasible distraction method for young children during needle procedures and is worthy of further study. However, our preliminary results did not demonstrate meaningful efficacy because the sample size was too small to show the efficacy of VR using a dome screen. A larger clinical trial is required to investigate and confirm the efficacy of VR using a dome screen as a distraction tool for young children during IV placement. Further developments in VR equipment and study processes are also needed.

## Supporting information

**S1 File. Consort 2010 checklist.**
(DOC)

**S2 File. Dataset.**
(XLSX)

**S3 File. Study protocol (English version).**
(DOCX)

**S4 File. Study protocol (Korean version).**
(DOCX)

## Author Contributions

**Conceptualization:** Woori Bae, Joong Wan Park, Jae Yun Jung, Do Kyun Kim, Young Ho Kwak.

**Data curation:** Ha Ni Lee, Joong Wan Park.

**Formal analysis:** Ha Ni Lee.

**Funding acquisition:** Joong Wan Park.

**Investigation:** Woori Bae, Joong Wan Park.

**Methodology:** Woori Bae, Joong Wan Park, Jae Yun Jung, Soyun Hwang, Do Kyun Kim, Young Ho Kwak.

**Project administration:** Woori Bae, Joong Wan Park.

**Resources:** Joong Wan Park.

**Supervision:** Joong Wan Park.

**Writing – original draft:** Ha Ni Lee.

**Writing – review & editing:** Ha Ni Lee, Joong Wan Park, Jae Yun Jung, Soyun Hwang, Do Kyun Kim, Young Ho Kwak.

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
