## [Decision Letter · Decision Letter 0]

1 Jul 2021

PONE-D-21-12619

Virtual Reality Environment Using a Dome Screen for Procedural Pain in Young Children during Intravenous Placement: A Pilot Randomized Controlled Trial

PLOS ONE

Dear Dr. Park,

Thank you for submitting your manuscript to PLOS ONE. After careful consideration, we feel that it has merit but does not fully meet PLOS ONE’s publication criteria as it currently stands. Therefore, we invite you to submit a revised version of the manuscript that addresses the points raised during the review process.

Please submit your revised manuscript by 11 august 2021. If you will need more time than this to complete your revisions, please reply to this message or contact the journal office at plosone@plos.org. Please include the following items when submitting your revised manuscript:

We look forward to receiving your revised manuscript.

Kind regards,

Prof. Prabath Nanayakkara, MD, PhD, FRCP

Academic Editor

PLOS ONE

Journal Requirements:

2.  Thank you for submitting your clinical trial to PLOS ONE and for providing the name of the registry and the registration number. The information in the registry entry suggests that your trial was registered after patient recruitment began. PLOS ONE strongly encourages authors to register all trials before recruiting the first participant in a study.

a) your reasons for your delay in registering this study (after enrolment of participants started);

b) confirmation that all related trials are registered by stating: “The authors confirm that all ongoing and related trials for this drug/intervention are registered”.

3. Please amend your manuscript to include your abstract after the title page.

Reviewers' comments:

Reviewer's Responses to Questions

**Comments to the Author**

1. Is the manuscript technically sound, and do the data support the conclusions?

Reviewer #1: Partly

Reviewer #2: Partly

2. Has the statistical analysis been performed appropriately and rigorously? 

Reviewer #1: No

Reviewer #2: N/A

3. Have the authors made all data underlying the findings in their manuscript fully available?

Reviewer #1: Yes

Reviewer #2: Yes

4. Is the manuscript presented in an intelligible fashion and written in standard English?

Reviewer #1: Yes

Reviewer #2: Yes

5. Review Comments to the Author

Reviewer #1: The objective of this randomized controlled trial (RCT) is to compare children (between 2-6 years) who were given IV placements to either the intervention (virtual reality using dome screen), or control groups. The study was registered as a RCT within the Korean trial registry system (with a valid KCT #), and was approved by the respective IRB/Ethics Committee. While the study objectives sound interesting, is important, and on target, some shortcomings were observed, in regards to abiding by the CONSORT guidelines for conducting and reporting results of high-quality randomized controlled trials (RCTs). Some other (statistical) comments were also provided.

1. Methods:

Methods reporting require an orderly manner following CONSORT guidelines, without repeating information, such as Trial Design, Participant Eligibility criteria and settings, Interventions, Outcomes, sample size/power considerations, Interim analysis and stopping rules, Randomization (details on random number generation, allocation concealment, implementation), Blinding considerations should be mentioned explicitly. The authors are advised to create separate subsections for each of the possible topics (whichever necessary), and that way produce a very clear writeup. I see the Authors already made a sincere attempt; however, they are advised to write it carefully, following nice examples in the manuscript below:

https://www.sciencedirect.com/science/article/pii/S0889540619300010

Specific comments:

(a) For instance, the randomization and allocation concealment should be made very clear (they are NOT the same thing); the trial staff recruiting patients should NOT have the randomization list. Randomization should be prepared by the trial statistician, and he/she would not participate in the recruiting.

(b) The randomization scheme employed here didn't appear to be very innovative; I would like to see a published reference (in this specific direction of research), where this ED-visit date based randomization was also considered. This can, very easily, lead to unbalanced groups. On the other hand, block randomization is often recommended in (pilot) trials, to ensure a balance in sample size across groups over time.

https://www.ncbi.nlm.nih.gov/pmc/articles/PMC2267325/

With the study already over, other randomizations couldn't be conducted further. However, I would like to see a clear justification on the pros and cons of this date-based randomization as a Discussion.

(c) Sample size/power: I am not necessarily agree that a sample size justification is not needed for a planning/pilot trial. See paper linked below. Can a post-hoc sample size computations be provided, so that readers have a proper idea on what they can expect, and/or what effect size the investigators wanted to base their study upon? Just choosing a number (like ~ 20 here) is not a well-thought-out science.

https://www.ncbi.nlm.nih.gov/pmc/articles/PMC4876429/

(d) Statistical Analysis: Overall, looks straightforward. Not clear, what's the point of using both independent t-tests, and Mann-Whitney U tests to assess group differences, wrt the endpoints. Be specific, why both, or one of the two procedures were used.

2. Results & Conclusions:

(a) The authors should check that any statement of significance should be followed by a p-value in the entire Results section. The Results section look OK.

(b) The Conclusions section should clearly state the very pilot nature of the study, using only 14 children. The results/conclusions are "only" from this (Korean) population, and allude to future studies with higher sample sizes, and/or combining other populations to determine the differences.

Reviewer #2: I would like to thank the authors for the opportunity to review the manuscript of their pilot study evaluating the use of a VR device in young children undergoing IV placement in the ED. It is a well written manuscript of a nicely done pilot RCT on a very interesting topic, however I have some remarks and some questions.

Abstract:

As the term PED is not used elsewhere in the abstract, I’d suggest leaving this abbreviation out.

I would recommend spelling out VR fully the first time it is used in the abstract and use the abbreviation subsequently.

Manuscript

Introduction:

Do you have a reference regarding the statement that needle procedures are the most common cause of pain in hospitalized children? (page 1, line 25)

Methods:

The authors identified their study as a pilot RCT. But on the other hand they state their objective as the aim “to investigate feasibility and acceptability of VR distraction..”. In general, a feasibility study aims to assess the feasibility of the intervention and will try to define endpoints, etc. in order to assess whether a full RCT might be feasible to conduct. On the other hand, a pilot RCT mainly assesses the processes for assessing eligibility, randomization and allocation and successful follow-up. Have the authors thought of the distinction between these two terms? A helpful reference had been published in 2011 by Abbott [PMID 25082389]. I’d suggest the authors to use this reference for specifying the objectives of the current study.

The study was registered beforehand in a trial register and the definitive study design and outcomes do not seem to differ from this published study protocol, decreasing the chances of publication bias. However, questions arise whether results such as FLACC scales, satisfaction and pain scales according to care givers can be adequately assessed in a pilot RCT setting, as there is no formal sample size calculation (which can be considered normal for a pilot RCT).

The complete intervention and standard care that was delivered to both groups has been well described. I’d specifically mention whether cutaneously applied local anesthetics were allowed while placing the IV or whether these were not utilized.

Allocation of treatment was not random, but was determined by date of presentation to the ED and therefore not concealed. Treatment was obviously not blinded to the patients (children and caregivers), however the authors state that the investigators were blinded while evaluating FLACC pain scores. How was this possible with the large dome screen in place?

Is it possible to supply the readers with a picture or photograph of the experimental setup?

Discussion

The most important limitation regarding conclusions drawn from the study is the small sample size. Although the authors mention this limitation, I’d suggest emphasizing this a bit more. On page 8, could you move the sentence in line 177 [Because of the …. using a dome screen] up to the beginning of this paragraph at line 174 to underscore the lack of statistical power?

6. PLOS authors have the option to publish the peer review history of their article (what does this mean?). If published, this will include your full peer review and any attached files.

Reviewer #1: No

Reviewer #2: **Yes: **Dr. M.L. Ridderikhof, MD PhD

---

## [Author Response · Author response to Decision Letter 0]

31 Jul 2021

Response to Journal Requirements

Response: Thank you for your comment. We revised the manuscript to conform to the style of PLOS ONE

2. Thank you for submitting your clinical trial to PLOS ONE and for providing the name of the registry and the registration number. The information in the registry entry suggests that your trial was registered after patient recruitment began. PLOS ONE strongly encourages authors to register all trials before recruiting the first participant in a study. As per the journal’s editorial policy, please include in the Methods section of your paper:

a) your reasons for your delay in registering this study (after enrolment of participants started);

Response: Thank you for your comment. Before starting this study, we believed that registration of the trial was not necessary because it was a pilot study. Therefore, this trial was registered only after the trial was completed to meet the international guidelines. We added some sentences about the reasons for delay in registering this study in the Methods section as follows.

“The institutional review board (IRB) of Seoul National University Hospital approved this study prior to its start date (IRB no. 1901-094-1005), and it was registered at cris.nih.go.kr (KCT0005691) after completion. Registration of this pilot trial was performed retrospectively because the authors were less aware of the required prospective registration for the pilot trial.”

b) confirmation that all related trials are registered by stating: “The authors confirm that all ongoing and related trials for this drug/intervention are registered”.

Response: Thank you for your comment. We added the sentence “The authors confirm that all ongoing and related trials for this drug/intervention are registered” in the Methods section as recommended.

3. Please amend your manuscript to include your abstract after the title page.

Response: Thank you for your comment. We added our abstract after the title page.

Response: Thank you for your comment. We added captions for our supporting information files at the end of the manuscript.

 

Response to Reviewers’ comments

Response: We truly thank the reviewers for their thoughtful comments about our manuscript. First, we would like to inform the reviewers that some errors in our manuscript were edited. There were some mistakes during the process of typing the calculated p values, and p values in Table 3 were incorrectly stated in the Results section of the previous manuscript. We edited the p values in Table 3. The significance of the values was not changed from previous results.

Reviewer #1:

The objective of this randomized controlled trial (RCT) is to compare children (between 2-6 years) who were given IV placements to either the intervention (virtual reality using dome screen), or control groups. The study was registered as a RCT within the Korean trial registry system (with a valid KCT #), and was approved by the respective IRB/Ethics Committee. While the study objectives sound interesting, is important, and on target, some shortcomings were observed, in regards to abiding by the CONSORT guidelines for conducting and reporting results of high-quality randomized controlled trials (RCTs). Some other (statistical) comments were also provided.

1. Methods:

Methods reporting require an orderly manner following CONSORT guidelines, without repeating information, such as Trial Design, Participant Eligibility criteria and settings, Interventions, Outcomes, sample size/power considerations, Interim analysis and stopping rules, Randomization (details on random number generation, allocation concealment, implementation), Blinding considerations should be mentioned explicitly. The authors are advised to create separate subsections for each of the possible topics (whichever necessary), and that way produce a very clear write up. I see the Authors already made a sincere attempt; however, they are advised to write it carefully, following nice examples in the manuscript below:

https://www.sciencedirect.com/science/article/pii/S0889540619300010

Response: Thank you for your comment. We edited the manuscript in an orderly manner and separated subsections following CONSORT guidelines as recommended.

Specific comments:

(a) For instance, the randomization and allocation concealment should be made very clear (they are NOT the same thing); the trial staff recruiting patients should NOT have the randomization list. Randomization should be prepared by the trial statistician, and he/she would not participate in the recruiting.

Response: Thank you for your comment. Participants in this pilot RCT were randomized according to the date of their ED visit because it was easy to implement. We should have selected a randomization method that minimizes the risk of bias, but we could not use this method. However, the baseline characteristics of the study groups were similar. Allocation concealment was also impossible for the trial staff recruiting patients because of this randomization method. We should use an appropriate randomization method, such as permuted block randomization, in future larger clinical trials. We described the randomization method and blinding in the Methods section as follows.

“Children were randomized to either the intervention group or the control group according to the dates of their ED visits. Children whose ED visit date was an odd number were assigned to the intervention group with VR distraction, and children whose ED visit date was an even number were assigned to the control group. Allocation concealment was impossible for the trial staff recruiting participants in this randomization method. Blinding was not feasible due to the nature of the intervention.”

We also added some sentences about the limitations of our randomization method in the Discussion section as follows.

“We should consider more appropriate randomization methods, such as permuted block randomization, in a future larger clinical trial to balance sample size across groups and minimize bias.”

(b) The randomization scheme employed here didn't appear to be very innovative; I would like to see a published reference (in this specific direction of research), where this ED-visit date based randomization was also considered. This can, very easily, lead to unbalanced groups. On the other hand, block randomization is often recommended in (pilot) trials, to ensure a balance in sample size across groups over time.

https://www.ncbi.nlm.nih.gov/pmc/articles/PMC2267325/

With the study already over, other randomizations couldn't be conducted further. However, I would like to see a clear justification on the pros and cons of this date-based randomization as a Discussion.

Response: Thank you for your comment. This randomization is not a good method, but we used it in this pilot study because of its easy implementation. A published reference is provided below, in which this ED visit date-based randomization was considered.

https://pubmed.ncbi.nlm.nih.gov/31661942/

Participants in our RCT were randomized according to the date of their ED visit. We chose this method because of its convenience in the pilot study, despite the high risk of selection bias. As we stated in the previous question, we should adopt an appropriate randomization method in a future clinical trial to minimize bias. We added some explanations about the randomization method in the Discussion section as follows.

“First, participants were randomized according to the date of their ED visit in this pilot study. We used this method for easy implementation. However, it may lead to unbalanced groups due to the high risk of selection bias. We should consider more appropriate randomization methods, such as permuted block randomization, in a future larger clinical trial to balance sample size across groups and minimize bias.”

(c) Sample size/power: I am not necessarily agree that a sample size justification is not needed for a planning/pilot trial. See paper linked below. Can post hoc sample size computations be provided so that readers have a proper idea on what they can expect and/or what effect size the investigators wanted to base their study upon? Just choosing a number (like ~ 20 here) is not a well-thought-out science.

https://www.ncbi.nlm.nih.gov/pmc/articles/PMC4876429/

Response: Thank you for your comment. We performed a post hoc sample size calculation using STATA version 14.2 and added some sentences about it in the Results section as follows.

“Based on these data in a post hoc sample size calculation (STATA version 14.2), we calculated that with 80% power and a 2-sided alpha of 0.05, a sample size of 36 per group would be required to detect possible subtle differences.”

(d) Statistical Analysis: Overall, looks straightforward. Not clear, what's the point of using both independent t-tests, and Mann-Whitney U tests to assess group differences, wrt the endpoints. Be specific, why both, or one of the two procedures were used.

Response: Thank you for your comment. Due to the small sample size, we used the Mann-Whitney U test to assess the efficacy of VR intervention. P values in Table 2 were also only from the Mann-Whitney U test. Although it is not necessary from a statistical point of view to use both tests, we wanted to show the numerical differences in pain scores between the groups in various ways. We edited Table 2 to clearly show the results.

2. Results & Conclusions:

(a) The authors should check that any statement of significance should be followed by a p-value in the entire Results section. The Results section look OK.

Response: Thank you for your comment. We checked the manuscript again as recommended.

(b) The Conclusions section should clearly state the very pilot nature of the study, using only 14 children. The results/conclusions are "only" from this (Korean) population and allude to future studies with higher sample sizes and/or combining other populations to determine the differences.

Response: Thank you for your comment. We added some sentences to the Conclusion section as follows.

“VR using a dome screen is a feasible distraction method for young children during needle procedures and is worthy of further study. However, our preliminary results did not demonstrate meaningful efficacy because the sample size was too small to show the efficacy of VR using a dome screen. A larger clinical trial is required to investigate and confirm the efficacy of VR using a dome screen as a distraction tool for young children during IV placement. Further developments in VR equipment and study processes are also needed.”

 

Reviewer #2:

I would like to thank the authors for the opportunity to review the manuscript of their pilot study evaluating the use of a VR device in young children undergoing IV placement in the ED. It is a well written manuscript of a nicely done pilot RCT on a very interesting topic, however I have some remarks and some questions.

Abstract:

As the term PED is not used elsewhere in the abstract, I’d suggest leaving this abbreviation out.

I would recommend spelling out VR fully the first time it is used in the abstract and use the abbreviation subsequently.

Response: Thank you for your comment. We edited the abstract as recommended.

“We assessed the feasibility and potential efficacy of a virtual reality (VR) environment using a dome screen as a distraction method in young children during intravenous (IV) placement in the pediatric emergency department.”

Manuscript

Introduction:

Do you have a reference regarding the statement that needle procedures are the most common cause of pain in hospitalized children? (page 1, line 25)

Response: Thank you for your comment. We added a reference for the statement that needle procedures are the most common cause of pain in hospitalized children.

https://onlinelibrary.wiley.com/doi/full/10.1111/aas.12846

Methods:

The authors identified their study as a pilot RCT. But on the other hand they state their objective as the aim “to investigate feasibility and acceptability of VR distraction.” In general, a feasibility study aims to assess the feasibility of the intervention and will try to define endpoints, etc. in order to assess whether a full RCT might be feasible to conduct. On the other hand, a pilot RCT mainly assesses the processes for assessing eligibility, randomization and allocation and successful follow-up. Have the authors thought of the distinction between these two terms? A helpful reference had been published in 2011 by Abbott [PMID 25082389]. I’d suggest the authors to use this reference for specifying the objectives of the current study.

Response: Thank you for your comment. The objectives of this clinical trial included the properties of a pilot study and a feasibility study. We specified the objectives of the study as follows in the Introduction section.

“The current pilot randomized clinical trial assessed the feasibility and acceptability of VR distraction using a dome screen for young children during needle procedures in PEDs. The secondary aims were to obtain preliminary results on the efficacy of VR distraction using a dome screen and to determine the sample size needed in a future larger clinical trial.”

The study was registered beforehand in a trial register, and the definitive study design and outcomes do not seem to differ from this published study protocol, decreasing the chances of publication bias. However, questions arise regarding whether results such as FLACC scales, satisfaction and pain scales according to caregivers can be adequately assessed in a pilot RCT setting, as there is no formal sample size calculation (which can be considered normal for a pilot RCT).

Response: Thank you for your comment. Because of the small sample size, this preliminary result may not have demonstrated the potential efficacy of VR. We could not reach the minimum sample size of 10 per arm as planned to perform this pilot study. A larger sample with precise sample size calculation should have been required to adequately test the efficacy of the intervention. Based on the results of this pilot study in a post hoc sample size calculation, we calculated that with 80% power and a 2-sided alpha of 0.05, a sample size of 36 per group would be required to detect possible subtle differences. We added some sentences about this in the Results section as follows.

“Based on these data in a post hoc sample size calculation (STATA version 14.2), we calculated that with 80% power and a 2-sided alpha of 0.05, a sample size of 36 per group would be required to detect possible subtle differences.”

The complete intervention and standard care that was delivered to both groups has been well described. I’d specifically mention whether cutaneously applied local anesthetics were allowed while placing the IV or whether these were not utilized.

Response: Thank you for your comment. We generally do not use local anesthetics for simple needle procedures in our hospital unless there is a caregiver’s request. We added some sentences to the Methods section as follows.

“No local anesthetics or analgesics other than the VR intervention were applied during the procedure.”

Allocation of treatment was not random, but was determined by date of presentation to the ED and therefore not concealed. Treatment was obviously not blinded to the patients (children and caregivers), however the authors state that the investigators were blinded while evaluating FLACC pain scores. How was this possible with the large dome screen in place?

Is it possible to supply the readers with a picture or photograph of the experimental setup?

Response: Thank you for your comment. The photo below is a captured scene from the video recording that was used for assessing the pain scale. We recorded video by adjusting the angle of view so that the dome screen was not visible in the scene. We also tried to blind the investigators to each child’s allocation as much as possible by setting the volume to zero in the video recording of both groups. However, as we mentioned in the Discussion section, blinding was not completely achieved because of the light of VR animation and the direction of children’s gaze toward the VR animation. We added some sentences in the Methods section as follows and added the photo named “Figure 2” in the Methods section to show our experimental setup to readers.

“By adjusting the angle of view, the dome screen was not visible in the recorded video to blind each participant’s allocation to PED staff who rated the pain scale from the recorded video.”

“The two PED staff members were blinded to each child’s allocation by setting the volume in the video recordings of both groups to zero and independently observing the video recording of the needle procedure for each child.”

Discussion

The most important limitation regarding conclusions drawn from the study is the small sample size. Although the authors mention this limitation, I’d suggest emphasizing this a bit more. On page 8, could you move the sentence in line 177 [Because of the …. using a dome screen] up to the beginning of this paragraph at line 174 to underscore the lack of statistical power?

Response: Thank you for your comment. To focus the limitations of the small number of participants, the paragraph in the Discussion section was edited as follows.

“Because of the small number of participants, it was impossible to show a statistically meaningful efficacy of VR distraction using a dome screen. The differences in the FLACC scale between the groups were not statistically significant overall, but the children allocated to the intervention group showed a numerically smaller average score on the FLACC scale. The guardians in the intervention group also showed great satisfaction with VR using a dome screen as a distraction method in terms of its effects on their children’s pain and anxiety.”

---

## [Editor Report · Decision Letter 1]

9 Aug 2021

Virtual reality environment using a dome screen for procedural pain in young children during intravenous placement: A pilot randomized controlled trial

PONE-D-21-12619R1

Dear Dr. Sir,

We’re pleased to inform you that your manuscript has been judged scientifically suitable for publication and will be formally accepted for publication once it meets all outstanding technical requirements.

Kind regards,

Prof, P W. B. Nanayakkara, MD, PhD, FRCP

Academic Editor

PLOS ONE

Additional Editor Comments (optional):

We are statisfied with their answers They have answered the question to the best of their ability.

Well done.

Reviewers' comments:

We are statisfied with their answers. They have answered the questions to the best of their ability.

Well done.

---

## [Editor Report · Acceptance letter]

23 Aug 2021

PONE-D-21-12619R1 

Virtual reality environment using a dome screen for procedural pain in young children during intravenous placement: A pilot randomized controlled trial 

Dear Dr. Park:

I'm pleased to inform you that your manuscript has been deemed suitable for publication in PLOS ONE. Congratulations! Your manuscript is now with our production department. 

Kind regards, 

on behalf of

Dr. P W. B. Nanayakkara 

Academic Editor

PLOS ONE